# Lipid-mediated Wnt protein stabilization enables serum-free culture of human organ stem cells

Nesrin Tüysüz[1], Louis van Bloois[2], Stieneke van den Brink[3,4], Harry Begthel[3,4], Monique M.A. Verstegen[5], Luis J. Cruz[6], Lijian Hui[7], Luc J.W. van der Laan[5], Jeroen de Jonge[5], Robert Vries[3,4], Eric Braakman[8], Enrico Mastrobattista[2], Jan J. Cornelissen[8], Hans Clevers[3,4] & Derk ten Berge[1]

Wnt signalling proteins are essential for culture of human organ stem cells in organoids, but most Wnt protein formulations are poorly active in serum-free media. Here we show that purified Wnt3a protein is ineffective because it rapidly loses activity in culture media due to its hydrophobic nature, and its solubilization requires a detergent, CHAPS (3-[(3-cholamidopropyl) dimethylammonio]-1-propanesulfonate), that interferes with stem cell self-renewal. By stabilizing the Wnt3a protein using phospholipids and cholesterol as carriers, we address both problems: Wnt activity remains stable in serum-free media, while non-toxic carriers allow the use of high Wnt concentrations. Stabilized Wnt3a supports strongly increased self-renewal of organ and embryonic stem cells and the serum-free establishment of human organoids from healthy and diseased intestine and liver. Moreover, the lipophilicity of Wnt3a protein greatly facilitates its purification. Our findings remove a major obstacle impeding clinical applications of adult stem cells and offer advantages for all cell culture uses of Wnt3a protein.

[1] Department of Cell Biology, Erasmus University Medical Center, PO Box 2040, Rotterdam 3000 CA, The Netherlands. [2] Department of Pharmaceutics, Utrecht Institute for Pharmaceutical Sciences, Utrecht University, Universiteitsweg 99, Utrecht 3584 CG, The Netherlands. [3] Hubrecht Institute, Royal Netherlands Academy of Arts and Sciences (KNAW), Cancer Genomics.nl and University Medical Center Utrecht, Uppsalalaan 8, Utrecht 3584 CT, The Netherlands. [4] Foundation Hubrecht Organoid Technology (HUB), Utrecht 3584 CT, The Netherlands. [5] Department of Surgery, Erasmus University Medical Center, PO Box 2040, Rotterdam 3000 CA, The Netherlands. [6] Experimental Molecular Imaging, Department of Radiology, Leiden University Medical Center, Albinusdreef 2, Leiden 2333 ZA, The Netherlands. [7] Shanghai Institute of Biochemistry and Cell Biology, Shanghai Institutes for Biological Sciences, Chinese Academy of Sciences, Shanghai 200031, China. [8] Department of Hematology, Erasmus University Medical Center, PO Box 2040, Rotterdam 3000 CA, The Netherlands. Correspondence and requests for materials should be addressed to D.t.B. (email: d.tenberge@erasmusmc.nl).

The recent establishment of organoid cultures from intestine, pancreas, liver and other human organs holds great promise for disease modelling, drug development, personalized medicine, and gene and stem cell therapies[1–5]. Organoids reproduce many organ properties, including disease symptoms and their response to therapeutics[6,7]. This allows the screening of drugs to select optimal treatments for, for example, cystic fibrosis[6] or colon cancer patients[7], bringing true personalized medicine to the patient. Self-renewal of the stem cells in the organoids requires activation of the Wnt pathway. In mouse organoids this is achieved by amplification of endogenous Wnt signals by the Wnt potentiator R-Spondin1 (ref. 1). In contrast, human organoids require additional Wnt ligands, provided by a serum-containing medium conditioned by a Wnt3a-producing cell line[3]. The conditioned medium contains undefined, differentiation-inducing components undesirable for diagnostic assays or other clinical applications. Moreover, screening of serum batches is necessary, and select sera support only some types of organoid, complicating culture. For diagnostic and therapeutic application, replacement of Wnt3a-conditioned media by purified factors would therefore be strongly preferred.

Wnt proteins are soluble signalling molecules that require attachment of a palmitoylate moiety to gain activity, and for this reason they are hydrophobic[8–10]. To maintain solubility, Wnt proteins are purified and stored in the presence of the detergent CHAPS (3-[(3-cholamidopropyl) dimethylammonio]-1-propane-sulfonate)[8]. However, on dilution in cell culture media, the detergent concentration drops below the level required to maintain Wnt solubility. This leads to rapid aggregation and loss of activity of the protein, in particular in the absence of serum[11]. Several studies have shown that Wnt proteins have a high affinity for phospholipid vesicles, likely due to their hydrophobicity[12,13], and it was recently shown that this association prolongs the activity of Wnt3a protein in the absence of serum[13].

In the current study, we found that purified Wnt3a protein performed poorly in the establishment and propagation of human organ stem cell cultures in serum-free conditions. We identified two factors responsible for this poor performance. First, insufficient Wnt activity is maintained due to the rapid loss of activity in serum-free medium. Second, the presence of CHAPS in the purified Wnt3a suppresses stem cell self-renewal. We demonstrate here that association of the hydrophobic Wnt3a protein with soluble lipid carriers, including liposomes and hydrophobic nanoparticles (NPs), enhances its stability such that it now supports organ stem cells in the absence of serum and CHAPS. Moreover, we show that the affinity of Wnt3a to lipids has applications in the purification of recombinant Wnt3a. Our findings constitute an important step towards the use of human organ stem cells in clinical scenarios.

## Results

**Purified Wnt3a protein adversely affects stem cell cultures.**
Adult human duodenum organoids were derived from intestinal biopsies as described[3]. However, while organoids were successfully derived using Wnt3a conditioned medium, we found that purified Wnt3a protein failed to support the derivation of duodenum organoids (Fig. 1a). Active Wnt proteins are palmitoylated[8–10] and require the detergent CHAPS to maintain solubility on purification[8]. On dilution in cell culture medium, the CHAPS concentration drops below the level required to maintain Wnt activity, and the protein rapidly loses activity[11]. To investigate whether activity loss of Wnt3a protein in serum-free medium caused its poor performance, we used the clonal expansion of mouse embryonic stem cells (ESCs) as a Wnt-sensitive stem cell assay[14]. Purified Wnt3a protein supported ESC self-renewal when added at every passage (3 days) (Fig. 1b), but daily addition was required when endogenous Wnt proteins were eliminated using the small-molecule inhibitor IWP2 (Fig. 1b), showing that purified Wnt3a protein provides only a short-lived stimulus. To determine its stability, we incubated Wnt3a protein for various periods of time in the culture medium at 37 °C and assayed the remaining activity using a luciferase reporter assay. While Wnt3a-conditioned medium retained activity over several days, purified Wnt3a lost its activity within a few hours (Fig. 1c). Surprisingly, when we doubled the concentration of Wnt3a to compensate for this rapid loss of activity, ESC self-renewal was repressed (Fig. 1d). This appeared due to a cytotoxic effect of the detergent CHAPS because doubling its concentration while maintaining the same level of Wnt3a repressed self-renewal to a similar extent (Fig. 1d). While CHAPS concentrations above 0.25% kill cells by lysing their membranes[13], CHAPS stayed below 0.02% in our experiments. This shows that low CHAPS concentrations that are readily accumulated during normal cell culture interfere with stem cell self-renewal. Thus, the use of purified Wnt3a protein suffers from dual impediments: it rapidly loses activity on addition to serum-free cell cultures, and the cytotoxicity of the detergent CHAPS prevents a compensating increase in Wnt3a concentration.

**Lipids enhance Wnt3a protein stability.** We and others previously observed that Wnt proteins associate with lipid vesicles, and that this prolongs their activity[12,13]. We explored whether liposomes would support Wnt3a solubility in the absence of CHAPS, and improve its ability to support stem cells. Liposomes were composed of the phospholipid DMPC (1,2-dimyristoyl-sn-glycero-3-phosphocholine), required to maintain Wnt3a activity[12]. Since DMPC liposomes rapidly aggregated and precipitated, we added the charged phospholipid DMPG (1,2-dimyristoyl-sn-glycero-3-phospho-rac-glycerol) to prevent aggregation. We further tested whether inclusion of cholesterol would enhance the physical stability of liposomes[15]. The liposomes were relatively uniformly sized, with diameters ranging from 130–150 nm and low polydispersity indices (Supplementary Table 1). On addition of Wnt3a protein to the liposomes, CHAPS was removed by dialysis (Supplementary Fig. 1a) with little effect on size distribution (Supplementary Table 1). About 85% of the Wnt3a protein associated with the liposomes (Supplementary Fig. 1b), in agreement with earlier measurements[12]. A dose–response assay showed that association with liposomes did not affect the specific activity of Wnt3a (Supplementary Fig. 1c). Importantly, the liposomes slowed the loss of Wnt3a activity over time, in particular at high levels of cholesterol (Fig. 2a), regardless of the removal of CHAPS (Fig. 2b). This holds promise for their use in serum-free stem cell cultures, and dialysed DMPC, DMPG and cholesterol (10:1:10 molar ratio) liposomes were therefore used in the remainder of this study.

**Lipid-stabilized Wnt3a supports serum-free stem cell culture.**
Wnt3a liposomes maintained the activity of the Wnt reporter 7xTcf-eGFP[16] for more than 3 days, while the activity started to decline 2 days after the addition of purified Wnt3a (Fig. 2c). Moreover, Wnt3a liposomes promoted a more than 10-fold higher expansion of undifferentiated ESCs (Fig. 2d), and higher proliferation of human duodenum organoid cells than purified Wnt3a (Fig. 2e). To specifically assay for the expansion of organ stem cells, we passaged the organoids twice at clonal density on dissociation into single cells and quantified the number of new organoids. While purified Wnt3a performed poorly in this assay,

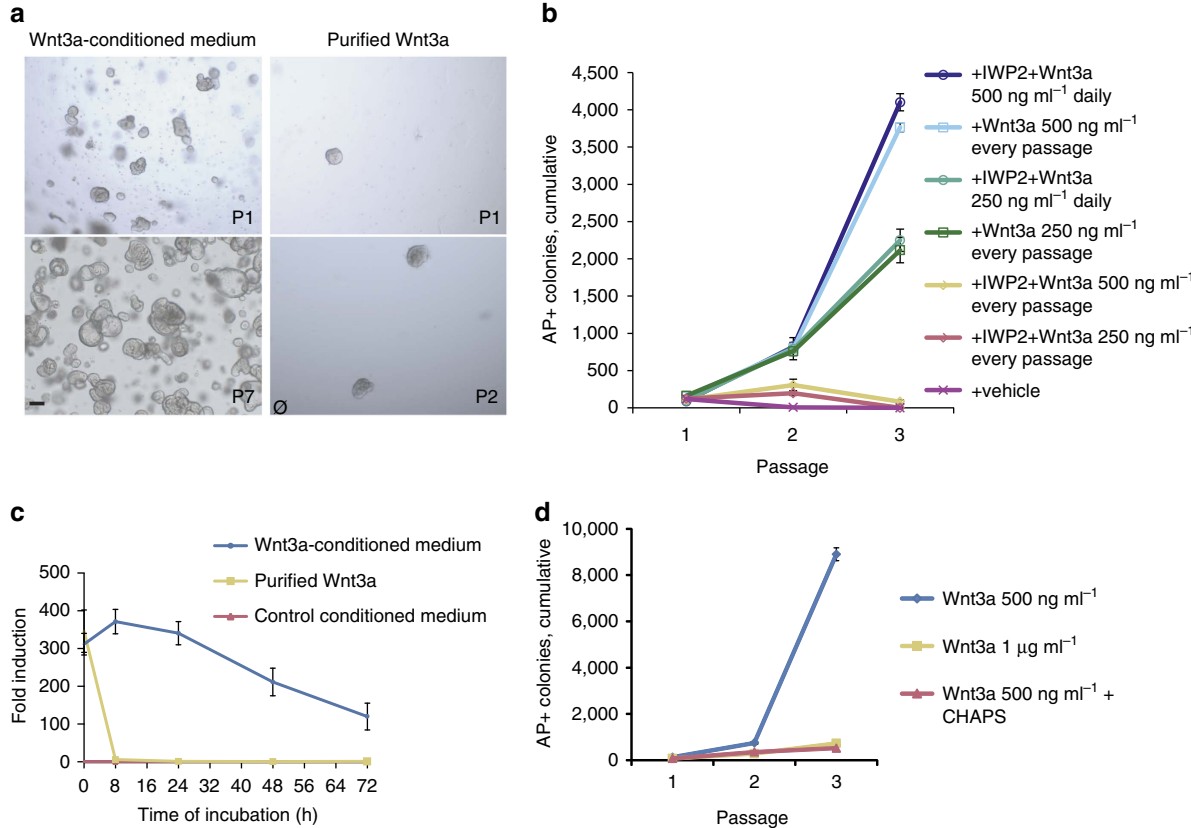

**Figure 1 | Instability and detergent-associated toxicity of Wnt3a protein adversely affect stem cell cultures.** (**a**) Human duodenum organoids derived in the presence of Wnt3a-conditioned medium, or purified Wnt3a protein ($400\,ng\,ml^{-1}$) in serum-free medium. (**b**) Expansion of ESCs able to form undifferentiated, alkaline phosphatase-positive (AP+) colonies at clonal density. Wnt3a and medium are refreshed daily or after every passage (3 days) ($n = 3$, mean ± s.e.m.). (**c**) Wnt activity retained after incubation for the indicated amounts of time at 37 °C. Wnt3a ($250\,ng\,ml^{-1}$) was diluted in serum-free medium ($n = 3$, mean ± s.e.m.). (**d**) Expansion of ESCs able to form AP+ colonies. Wnt3a and media are refreshed daily ($n = 3$, mean ± s.e.m.). Scale bar, 100 µm. P, passage.

Wnt3a liposomes efficiently supported organoid formation from single cells in a dose-dependent manner (Fig. 3a). Raising the concentration of purified Wnt3a ultimately suppressed self-renewal (Fig. 3a), suggesting that here too CHAPS inhibited self-renewal. Indeed, Wnt3a liposomes, lacking CHAPS, strongly promoted stem cell expansion at concentrations over $1\,µg\,ml^{-1}$ Wnt3a (Fig. 3a). Moreover, and in contrast to regular purified Wnt3a, Wnt3a liposomes supported efficient *de novo* derivation and long-term maintenance of duodenum organoids in serum-free medium (Figs 1a and 3b). The newly derived duodenum organoids displayed robust long-term expansion and could be maintained in Wnt3a liposomes for more than 6 months at passaging ratios of 1:6–1:8 every 7–10 days. These data show that dialysed Wnt3a liposomes promoted stem cell self-renewal by eliminating the cytotoxic CHAPS from the cultures while prolonging Wnt activity.

We verified that the newly derived duodenum organoids expressed stem cell, proliferation and differentiation markers as found in the intestinal crypt, and that their expression was comparable with organoids derived in the presence of Wnt3a conditioned medium. The intestinal stem cell markers LGR5 (ref. 17) and TROY[18] were expressed at similar level as in organoids derived in the Wnt3a conditioned medium (Fig. 3c), suggesting that Wnt3a liposomes and the conditioned medium maintained similar proportions of stem cells in the organoids. The tyrosine kinase receptor EPHB2 is highly expressed in intestinal crypts[1,19] (Fig. 3d, arrows) and immunostaining revealed high expression throughout the organoids (Fig. 3d). Likewise, expression of the

proliferation marker Ki67 was found in the crypts and throughout the organoids (Fig. 3d, arrows). Differentiation markers for enterocytes (alkaline phosphatase), goblet cells (periodic acid-Schiff staining) and endocrine cells (chromogranin A) were absent from both the crypts and the organoids, while clearly detectable in the differentiated intestinal epithelium (Fig. 3d, arrows). These data show that organoids derived in the presence of Wnt3a liposomes display characteristics of the proliferative stem cell compartment of the intestinal crypt. Moreover, we verified the multilineage potential of the organoids by inducing their differentiation towards enterocytes, goblet cells and enteroendocrine cells. On removal of liposomal Wnt3a, cells carrying the enterocyte marker Villin appeared within 5 days (Fig. 3e). Differentiation towards goblet and enteroendocrine cells was induced by the removal of SB202190 and nicotinamide[3]. Expression of the goblet marker Mucin2 and the enteroendocrine marker chromogranin A was visible within 5 days of induction (Fig. 3e). Moreover, in line with previous findings[3], the Paneth cell marker lysozyme was present both in expansion and differentiation conditions, indicating the presence of Paneth cells in the organoids (Fig. 3e). Together, these data show that the organoids derived in Wnt3a liposomes contained multipotent stem cells and were similar to those derived in Wnt3a-conditioned medium with regard to cell types present and similarity to the intestinal crypt.

Palmitoylated proteins reversibly interact with lipid bilayers[20], which may underlie the interaction of Wnt3a with liposomes. We tested whether the prior association of Wnt protein with

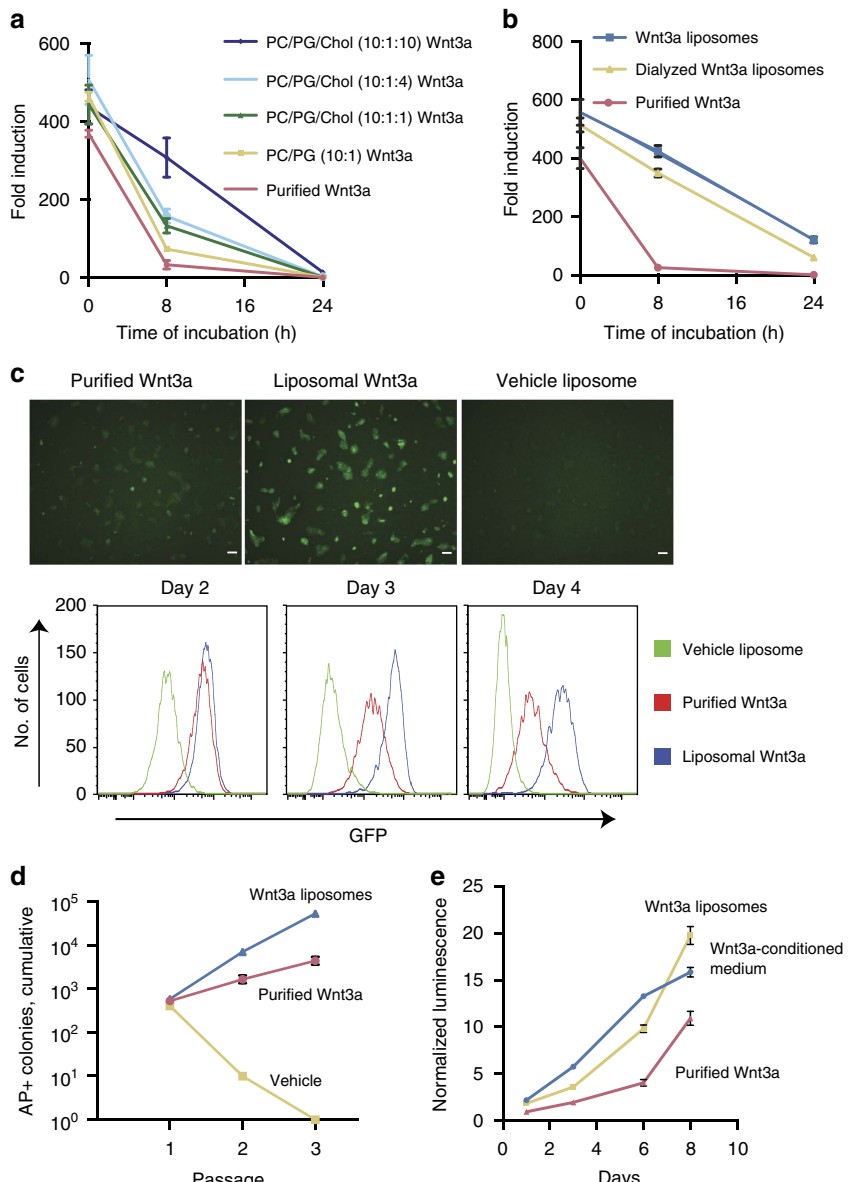

**Figure 2 | Lipid-stabilized Wnt3a protein shows enhanced target gene activation and embryonic stem cell expansion in serum-free culture.**
(**a**) Quantification of Wnt activity retained after incubation of the indicated liposomes (containing 250 ng ml$^{-1}$ Wnt3a final concentration) in serum-free medium for the indicated amounts of time at 37 °C ($n = 3$, mean ± s.e.m.). (**b**) Wnt activity retained after incubation of DMPC/DMPG/Chol 10:1:10 liposomes in serum-free medium for the indicated amounts of time at 37 °C ($n = 3$, mean ± s.e.m.). (**c**) ESCs carrying the Wnt reporter 7xTcf-eGFP were imaged for GFP 3 days after the addition of purified or liposomal Wnt3a (250 ng ml$^{-1}$) in serum-free medium (upper panels) and analysed by flow cytometry over 4 days (lower panels). (**d**) Quantification of expansion of R1 ESCs able to form alkaline phosphate-positive (AP +) colonies at clonal density in serum-free medium. Indicated reagents are added after every passage (3 days) ($n = 3$, mean ± s.e.m.). (**e**) Expansion of human duodenum organoids cultured with the indicated reagents, determined by ATP assay ($n = 3$, mean ± s.e.m.). Scale bars, (**c**) 100 µm.

liposomes was required for the stabilizing effect in cell culture. However, separate addition of liposomes also prolonged the activity of Wnt3a in a lipid-concentration-dependent manner (Fig. 4a). Moreover, separate addition of liposomes and purified Wnt3a supported the clonal expansion of duodenum stem cells (Fig. 4b). This shows that simple addition of liposomes to organoid medium enhances the stability of Wnt3a protein and allows the serum-free expansion of human organoids. We explored alternatives for liposomes such as hydrophobic NPs that can be more easily stored and shipped. Poly(D, L-lactide-co-glycolide) (PLGA) is an FDA-approved biodegradable copolymer with excellent biocompatibility properties and long shelf-life[21].

We coated PLGA NPs with DMPC, stored them for 1 year at 4 °C to test the shelf-life and incubated them with Wnt3a protein, followed by dialysis to remove CHAPS. The NPs stabilized Wnt3a activity to a similar degree as liposomes (Fig. 4c). Moreover, separate addition of Wnt3a protein and DMPC-coated NPs also prolonged Wnt activity (Fig. 4c). Thus, Wnt3a activity in serum-free cell cultures can easily be prolonged by the addition of PLGA NPs. However, the presence of CHAPS limits the concentration of Wnt3a that can be used via this approach. To obtain a CHAPS-free stabilized Wnt3a reagent with long shelf-life, we lyophilized dialyzed Wnt3a liposomes. On reconstitution, lyophilized Wnt3a liposomes retained most of their activity and physical properties

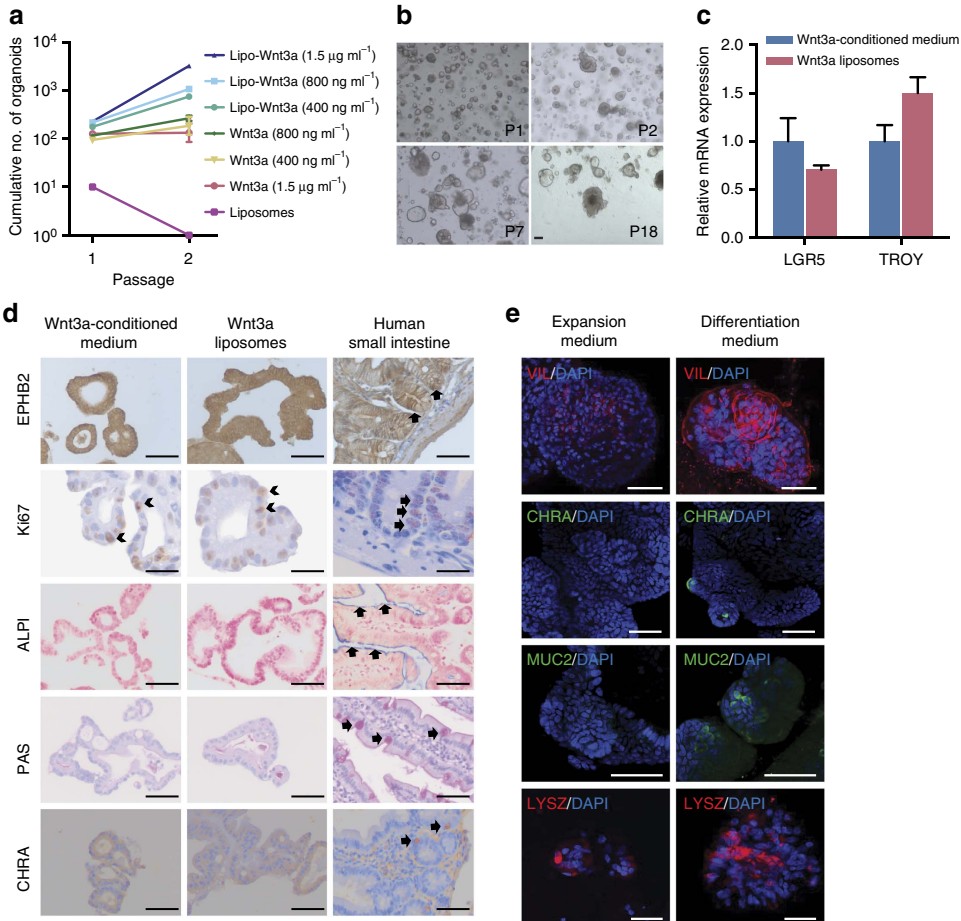

**Figure 3 | Lipid-stabilized Wnt3a protein supports serum-free derivation and self-renewal of multipotent human intestinal stem cells.** (**a**) Expansion of single human duodenum organoid cells (passage 7) able to form new organoids ($n = 3$, mean ± s.e.m.). (**b**) Human duodenum organoids derived in the presence of Wnt3a liposomes (400 ng Wnt3a per ml) in serum-free medium. (**c**) Quantitative RT–PCR analysis for the intestinal stem cell markers LGR5 and TROY in passage 7 human duodenum organoids derived and maintained in Wnt3a-conditioned medium or liposomes ($n = 3$, mean ± s.e.m.). (**d**) Histochemical and immunocytochemical stainings as indicated of sections of passage 6 human duodenum organoids derived and maintained in Wnt3a-conditioned medium or liposomes, and of human small intestine. Staining for Ki67, EPHB2 and chromogranin A (CHRA) in brown, for alkaline phosphatase (ALPI) in blue and periodic acid-Schiff (PAS) in purple (see arrows). (**e**) Confocal images (z-stack projection) of human duodenum organoids derived and maintained in Wnt3a liposomes, then either differentiated for 5 days or maintained in expansion medium. Immunostaining for Villin (VIL) and lysozyme (LYSZ) in red, and for Mucin2 (MUC2) and CHRA in green. Nuclei were counterstained with DAPI (4,6-diamidino-2-phenylindole; blue). Scale bars, (**b**) 100 μm, (**d**) 20 μm (Ki67) and (**d,e**) 50 μm.

(Fig. 4d and Supplementary Table 2), thereby providing Wnt3a protein in a stabilized CHAPS-free storable format that is generally applicable and supports the serum-free derivation and maintenance of human organoid cultures. Thus, we developed several formats for enhancing Wnt3a stability in serum-free cell culture, depending on the sensitivity of the cell culture system to CHAPS and the final Wnt3a concentration required.

Since recombinant Wnt proteins are purified from conditioned media, they contain serum-derived contaminants. While low levels of these contaminants are compatible with clinical applications, as long as such sera are verified free from known infectious disease markers (for example, bovine sera sourced from certified transmissible spongiform encephalopathy/bovine spongiform encephalopathy-free herds), they may affect stem cell cultures. We found that highly purified Wnt3a, of more than 90% purity, demonstrated much lower stability than less pure preparations (Fig. 5a,b). While this suggests that some contaminants enhance Wnt protein stability, the high-purity Wnt3a was nonetheless successfully stabilized by liposomes (Fig. 5b) and supported the clonal expansion of human duodenum stem cells

(Fig. 5c). The development of serum-free stabilizers to produce Wnt proteins or of alternative molecules that activate the Wnt pathway are potential avenues towards completely eliminating sera. Recently, it was shown that the serum glycoprotein afamin forms a complex with Wnt3a that remains soluble in aqueous buffer[22]. However, we found that recombinant afamin was unable to prolong the activity of purified Wnt3a when present at concentrations found in serum-containing media[23] (Supplementary Fig. 3). Possibly, recombinant afamin lacks essential modifications or its stabilization of Wnt3a only occurs when the molecules are complexed during their biosynthesis. Glycogen synthase kinase 3 inhibitors such as CHIR99021 support strong β-catenin stabilization and are far cheaper, more stable and easier to use than Wnt ligands[24]. However, glycogen synthase kinase 3 inhibitors cannot always substitute for Wnt ligands in the maintenance of stem cells[25], and we were unable to propagate human duodenum organoids using CHIR99021 in lieu of purified Wnt3a (Supplementary Fig. 4). Since purifying Wnt3a to very high purity lowers yields and thus raises costs, we explored whether the affinity of Wnt3a for liposomes could be exploited to

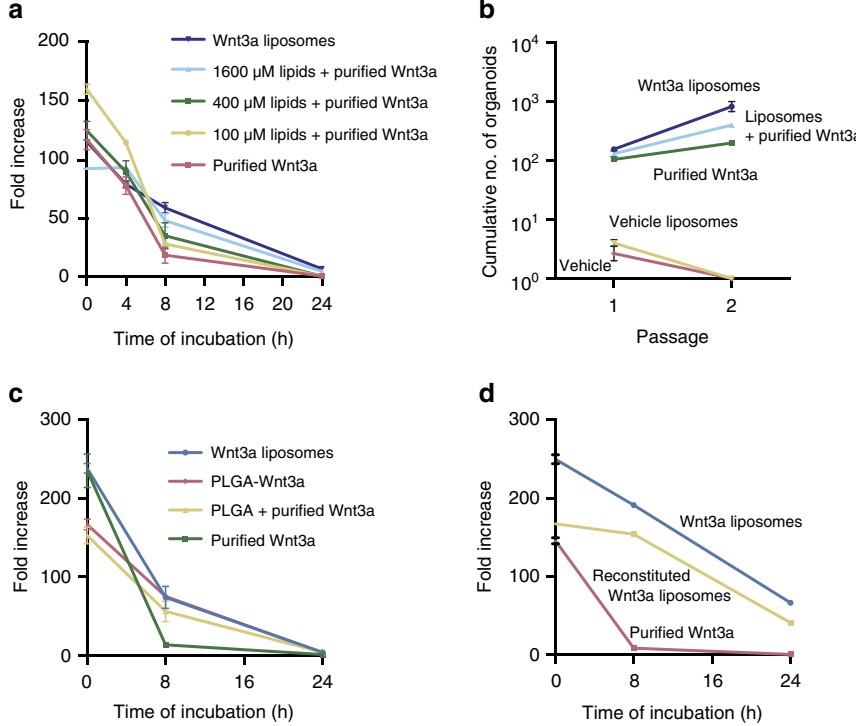

**Figure 4 | Alternative lipid carriers stabilize Wnt3a protein.** (**a**) Quantification of Wnt activity retained after incubation of 250 ng ml$^{-1}$ Wnt3a protein in the indicated conditions in serum-free medium for the indicated amounts of time at 37 °C. Liposomes and purified Wnt3a protein were added separately, except for 'Wnt3a liposomes' ($n = 3$, mean ± s.e.m.). (**b**) Expansion of single human duodenum organoid cells (passage 10) able to form new organoids in the presence of Wnt3a (800 ng ml$^{-1}$) ($n = 3$, mean ± s.e.m.). (**c**) Wnt3a activity retained after incubation for the indicated amounts of time in serum-free medium at 37 °C ($n = 3$, mean ± s.e.m.). DMPC-coated PLGA nanoparticles were preincubated with Wnt3a (PLGA-Wnt3a) or added simultaneously with Wnt3a (PLGA + purified Wnt3a). (**d**) Wnt activity retained after incubation for the indicated amounts of time at 37 °C (250 ng ml$^{-1}$ Wnt3a).

remove contaminants from Wnt3a preparations. Indeed, by separating the liposomes from a suspension of low-purity Wnt3a liposomes, we recovered more than 80% of Wnt3a protein while almost all contaminants were removed, leaving only BSA as a significant presence (Supplementary Fig. 1b and Fig. 5d). These data show that liposomes facilitate not only the stabilization but also the production of high-purity Wnt3a protein.

Finally, we explored whether Wnt3a liposomes would support the derivation of stem cell cultures from other human organs. Indeed, Wnt3a liposomes supported organoid derivation from jejunum biopsies (Fig. 6a, Table 1 and Supplementary Fig. 2a,b), unlike purified Wnt3a protein and, interestingly, unlike Wnt3a-conditioned medium (Table 1 and Supplementary Fig. 2a). This underscores the inconsistent nature of the Wnt3a-conditioned medium, which was prepared using duodenum-screened serum, and highlights an additional benefit of Wnt3a liposomes as a more universally applicable reagent. Jejunum organoids were maintained for more than 3 months and displayed robust expansion in the presence of Wnt3a liposomes (Supplementary Fig. 2b). Furthermore, Wnt3a liposomes supported the derivation of human liver organoids[26] from both three out of four healthy donors and from three cases with end-stage liver disease (Fig. 6b and Table 2). Wnt liposomes hold therefore promise for modelling liver diseases and may have wider applicability.

## Discussion

Our study shows that the main technical obstacles with using Wnt protein for serum-free stem cell cultures are its instability and its detergent-associated stem cell toxicity. The prolonged activity of lipid-stabilized Wnt3a and the absence of CHAPS

advance an approach to establish defined long-term cultures of organoids from human organ stem cells. This removes an obstacle for the use of these cells in clinical scenarios, and lipid-stabilized Wnt3a holds therefore considerable translational potential. Moreover, the lipid stabilization is compatible with high concentrations of Wnt3a protein and offers advantages for all tissue culture uses of Wnt3a protein. This is demonstrated by the considerably improved performance of Wnt3a liposomes in embryonic stem cell culture.

Several approaches have been used to increase the stability and activity of Wnt3a in serum-free conditions. In one of the first attempts, serum was fractionated and heparan sulfate proteoglycans (HSPGs) identified as Wnt-stabilizing components[11]. Due to their high cost, purified HSPGs are not currently economically viable in cell culture applications. Moreover, the basement membrane extract in which the organ stem cells are cultured is already rich in HSPGs[27]. Recently, the serum glycoprotein afamin was identified as another serum component able to stabilize Wnt3a[22]. Recombinant afamin did, however, not prolong the activity of purified Wnt3a in our hands. Since its stabilizing ability was demonstrated in a coexpression setting[22], this suggests that it depends on the molecules interacting during their biosynthesis or downstream cellular processing. Moreover, this suggests that yet other serum components must be able to stabilize Wnt3a on its dilution in serum-containing medium. This is also indicated by our finding that serum-derived contaminants substantially contribute to Wnt stability, given the inverse correlation that we observed between Wnt3a purity and stability.

Our work adds several other tools to the Wnt instrumentarium: Lipid-coated PLGA NPs can be stored for years[28], and we show that such particles are as effective as liposomes in

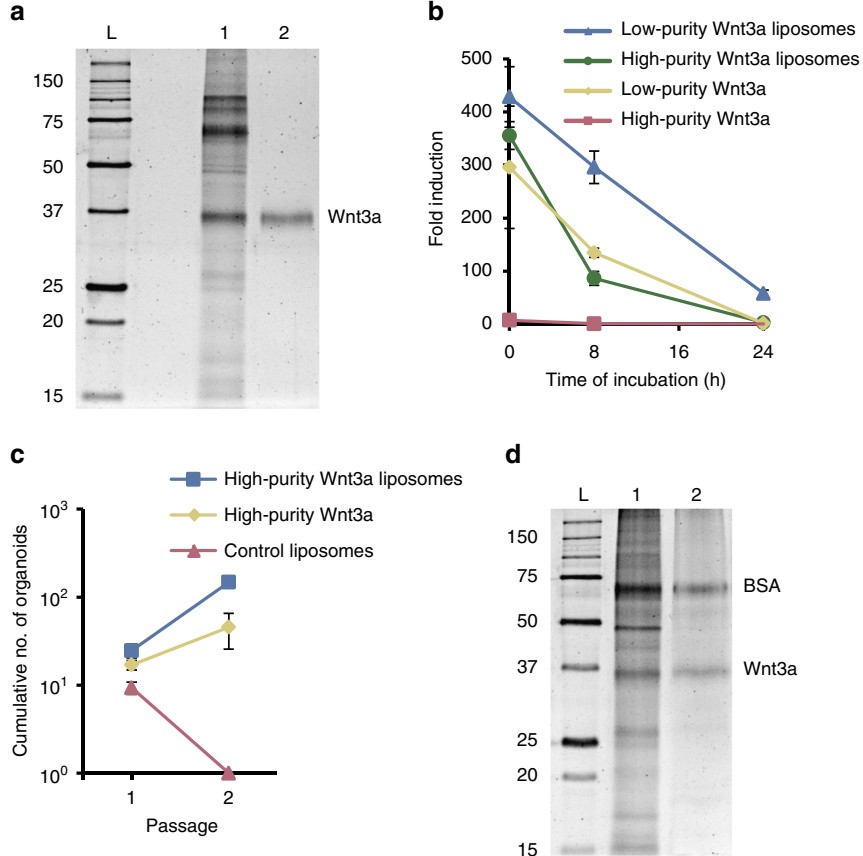

**Figure 5 | Liposome-mediated Wnt3a purification.** (**a**) SDS–polyacrylamide gel electrophoresis (SDS–PAGE) analysis of batches of Wnt3a protein of low (1) and high purity (2) detected by Oriole stain. (**b**) Quantification of Wnt activity retained after incubation of 250 ng ml$^{-1}$ Wnt3a protein from the batches of (**a**) in the indicated conditions in serum-free medium for the indicated amounts of time at 37 °C ($n=3$, mean ± s.e.m.). (**c**) Expansion of single human duodenum organoid cells (passage 9) able to form new organoids in the presence of 800 ng ml$^{-1}$ high-purity Wnt3a protein as indicated ($n=3$, mean ± s.e.m.). (**d**) SDS–PAGE analysis of low-purity Wnt3a liposome suspension (1) and the same liposomes after separation from the suspension (2), detected by Oriole stain. L, molecular weight ladder (kDa).

prolonging the activity of Wnt3a. Alternatively, for systems that require the elimination of CHAPS, we show that freeze-dried Wnt3a liposomes retain their activity on reconstitution. Finally, liposomes efficiently recover Wnt3a protein from crude preparations while leaving behind most contaminants, which may be applied to improve the economics of Wnt protein purification. Together, these technologies facilitate distribution of and access to stabilized Wnt ligands.

## Methods

**Culture of human intestinal and liver organoids.** Intestinal biopsies were obtained from patients with informed consent from Wilhelmina Children's Hospital, Utrecht and Erasmus Medical Center, Rotterdam. The biopsies were included on availability and none were excluded. The use of donor materials for research purposes was approved by the Medisch Ethische Toetsings Commissies (Medical Ethics Committees) of Utrecht Medical Center and of Erasmus Medical Center, and informed consent was obtained from all subjects. Intestinal samples were collected, washed with cold PBS until the supernatant was clear and tissue fragments incubated with 2 mM EDTA in cold chelation buffer (5.6 mM Na$_2$HPO$_4$, 8.0 mM KH$_2$PO$_4$, 96.2 mM NaCl, 1.6 mM KCl, 43.4 mM sucrose, 54.9 mM D-sorbitol, 0.5 mM DL-dithiothreitol) for 30 min on ice. Tissue fragments were vigorously resuspended in cold chelation buffer using a 10 ml pipette to isolate intestinal crypts. After settling of tissue fragments, the supernatant containing crypts was collected, the crypts pelleted, washed with cold chelation buffer and centrifuged at 150$g$ to remove single cells. Crypts were then mixed with Cultrex Basement Membrane Extract, Type 2 (BME; Amsbio) on ice and 50 μl drops plated in a 24-well plate (1,000 crypts per fragments per 50 μl of BME per well). The plate was incubated for 10 min at 37 °C to allow the BME to solidify, after which 500 μl of warm complete culture medium was overlaid. Outgrowing crypts of human duodenum and jejunum were typically refreshed every other day.

Complete culture medium consisted of basal culture medium (Advanced DMEM/F12 supplemented with 100 U ml$^{-1}$ penicillin and 100 μg ml$^{-1}$ streptomycin, 10 mM HEPES and glutamax (all from Invitrogen)) supplemented with 1 × B27 (Invitrogen), 1.25 mM N-acetyl cysteine (Sigma), 10 mM nicotinamide, 500 nM A83-01 (Tocris), 10 μM SB202190 (Sigma), 50 ng ml$^{-1}$ epidermal growth factor (Invitrogen), 1 μg ml$^{-1}$ Rspo1 (Nuvelo) or 20% Rspo1-conditioned medium and 100 ng ml$^{-1}$ Noggin (Peprotech) or 10% Noggin-conditioned medium, and 400–800 ng ml$^{-1}$ purified or liposomal Wnt3a or 50% Wnt3a-conditioned medium.

Rspo1- and Noggin-conditioned media were produced by conditioning basal culture medium for 1 week using HEK293 cells stably transfected with HA-mouse-Rpso1-Fc (gift from Calvin Kuo, Stanford University) or mouse Noggin-Fc expression vector[29], respectively. Wnt3a-conditioned medium was produced by conditioning medium containing 10% screened fetal bovine serum (Sigma) for 1 week using Wnt3a-expressing L cells (gift from Roel Nusse, Stanford University). All cells lines were tested for mycoplasma contamination every 2–3 months. Human or mouse Wnt3a protein was purchased from R&D Systems, or produced in Drosophila S2 cells grown in suspension culture (gift from Roel Nusse, Stanford University) and purified using Blue Sepharose affinity and gel filtration chromatography. For this, the S2 cells were expanded in Schneider's Drosophila medium (Lonza) containing 10% fetal bovine serum and antibiotics, and media were collected when cell expansion reached a plateau. Up to 12 L of conditioned medium was 0.45 μm filtered, adjusted to 1% Triton X-100 and applied to an FPLC column containing 200 ml Blue Sepharose 6 Fast Flow (GE Healthcare; 17094801). After washing with four volumes of washing buffer (150 mM KCl, 20 mM Tris-HCl, 1% CHAPS, pH 7.5), bound proteins were eluted with elution buffer (1.5 M KCl, 20 mM Tris-HCl, 1% CHAPS, pH 7.5). After analysis on a Coomassie gel, Wnt3a-containing fractions were selected, combined and concentrated to 10 ml using Pierce Concentrators 20K MWCO (Thermo Scientific; 89887A). The combined fractions were fractionated on a HiLoad 26/60 Superdex 200 gel filtration column (GE Healthcare) in PBS, 0.5 M NaCl, 1% CHAPS, pH 7.3. Fractions were analysed for purity and Wnt activity by Coomassie gel and Wnt activity assay (see below), and selected fractions were combined for use.

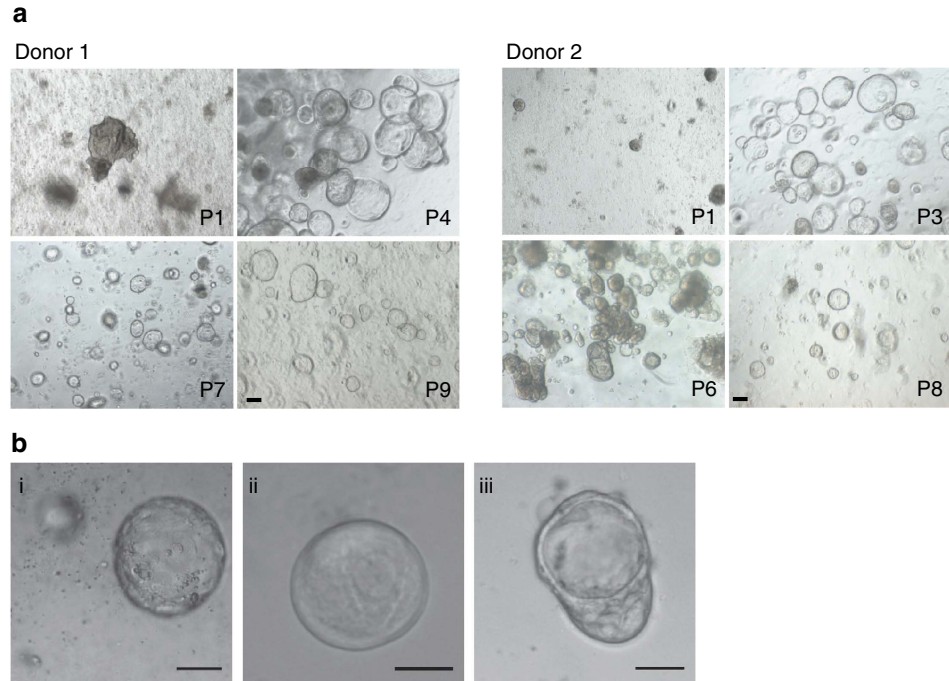

**Figure 6 | Wnt3a liposomes support serum-free establishment of human jejunum and liver organoids.** (**a**) Human jejunum organoids derived in the presence of Wnt3a liposomes (800 ng Wnt3a per ml) in serum-free medium. (**b**) Human liver organoids derived in the presence of Wnt3a liposomes (800 ng Wnt3a per ml) in serum-free medium from patient biopsies with (i) hepatitis C combined with hepatocellular carcinoma, (ii) α-1 antitrypsin deficiency and (iii) Wilson's disease. Scale bars, (**a**) 100 μm and (**b**) 50 μm. P, passage.

**Table 1 | Wnt3a liposomes support serum-free derivation of human small intestinal organoids.**

| Wnt3a reagent | Concentration | Donor and tissue type | | |
|---|---|---|---|---|
| | | Duodenum | Donor 1 Jejunum | Donor 2 Jejunum |
| Wnt3a CM | 50% | ✓ | — | — |
| Purified Wnt3a | 200 ng ml$^{-1}$ | ND | — | ND |
| Purified Wnt3a | 400 ng ml$^{-1}$ | — | — | — |
| Purified Wnt3a | 800 ng ml$^{-1}$ | ND | — | — |
| Purified Wnt3a | 1,000 ng ml$^{-1}$ | ND | ND | — |
| Vehicle liposomes | 0 ng ml$^{-1}$ Wnt3a | — | — | — |
| Wnt3a liposomes | 200 ng ml$^{-1}$ | ND | — | ND |
| Wnt3a liposomes | 400 ng ml$^{-1}$ | ✓ | — | ✓ |
| Wnt3a liposomes | 800 ng ml$^{-1}$ | ✓ | ✓ | ✓ |
| Wnt3a liposomes | 1,000 ng ml$^{-1}$ | ND | ND | ✓ |

CM, conditioned medium; ✓, successful organoid establishment; ND, not determined.
Derivations of organoids from human duodenum and jejunum biopsies in the presence of the indicated Wnt3a reagents.

**Table 2 | Wnt3a liposomes support serum-free derivation of human liver organoids.**

| Human liver tissue source | Underlying disease | Number of donors | Successful organoid derivations |
|---|---|---|---|
| Multi-organ donors | Cardiac arrest and/or brain death | 4 | 3 |
| End-stage liver disease patients | HCV combined with HCC | 1 | 1 |
| | α-1 Antitrypsin deficiency | 1 | 1 |
| | Wilson's disease | 1 | 1 |

HCC, hepatocellular carcinoma; HCV, hepatitis C virus.
Derivations of organoids from biopsies of human liver obtained from multi-organ donors and end-stage liver disease patients in the presence of Wnt3a liposomes (800 ng Wnt3a per ml).

Human liver biopsies (0.5–1 cm$^3$) were obtained from multi-organ donors and explant livers during liver transplantations performed at the Erasmus MC, Rotterdam. The biopsies were included on availability and none were excluded. The use of both donor and recipient materials for research purposes was approved by the Medisch Ethische Toetsings Commissie (Medical Ethics Committee) of Erasmus Medical Center, and informed consent was obtained from all subjects. To isolate liver cells from the biopsies, the minced tissue was washed with DMEM/1% FCS and digested with collagenase D (2.5 mg ml$^{-1}$, Roche) and DNase I (0.1 mg ml$^{-1}$, Sigma) in Earle's balanced salt solution (Thermoscientific) for 30 min at 37 °C. After the addition of cold DMEM/1% FCS, the suspension was filtered through a 70 μm strainer and pelleted for 5 min at 300$g$. The material retained in the strainer was further digested for 10 min with Accutase (Gibco) at 37 °C and strained again. The strained fractions were combined, washed with cold Advanced DMEM/F12 and pelleted at 300$g$ for 5 min. The cell pellet was mixed

with Matrigel (BD Biosciences) and 10,000 cells were seeded per well in a 48-well plate. The cell–Matrigel mix was incubated for 30 min at 37 °C, after which 250 µl culture medium was added. Culture media consisted of Advanced DMEM/F12 (Invitrogen) supplemented with N2 and B27 without retinoic acid (Gibco), 1.25 mM N-acetyl cysteine (Sigma), 10 nM gastrin (Sigma), 50 ng ml$^{-1}$ epidermal growth factor (Peprotech), 10% RSPO1 conditioned medium, 100 ng ml$^{-1}$ FGF10 (Peprotech), 25 ng ml$^{-1}$ HGF, 10 mM nicotinamide (Sigma), 5 µM A83.01 (Tocris) and 10 µM FSK (Tocris). For the establishment of the culture, the first 3 days after isolation the medium was supplemented with 25 ng ml$^{-1}$ Noggin (Peprotech), Wnt3a liposomes (800 ng ml$^{-1}$) and 10 µM Y27632 (Sigma).

**Intestinal organoid proliferation assay.** To quantify proliferation rate, organoid cultures were dissociated into single cells by incubating with TrypLE reagent (Invitrogen) at 37 °C for 5–10 min. Viable cells were counted and 1,500–3,500 viable single cells were seeded with 100 µl of complete culture medium supplemented with either Wnt CM or liposomal Wnt3a into U-bottom 96-well plates in five replicates for each condition. The cultures were incubated for 1, 3, 6 and 8 days. The culture medium was then removed from the wells and 50 µl of basal medium was added. Next, an equal volume of CellTiter-Glo Reagent (Promega) was added directly to the wells. Plates were incubated at room temperature for 10 min on a shaker in the dark. The samples were transferred into opaque-walled white 96-well plates for luminescence measurements done on a Centro XS LB 960 Multiplate Luminometer (Berthold Technologies). Values were normalized and expressed as the mean of three wells ± s.e.m. relative to the starting cell numbers.

**Clonal organoid formation assay.** For quantification of organoid-forming cells, organoids cultured were dissociated into single cells by TrypLE express (Life Technologies), and 2,000 single cells mixed with BME (50 µl per well) were seeded into 24-well plates in triplicates. The single cells were cultured in complete culture medium (as described above) containing the indicated concentration of purified or liposomal Wnt3a or CHIR99021 (Tocris; 4423-10). Media were changed every other day. After 7–10 days, the number of organoids formed was counted. The organoids were again dissociated into single cells and passaged at a dilution that would result in 2,000 single cells per well again. The cumulative number of organoids formed at this passage was determined after correcting for the dilution factor used during passaging. Results were plotted as the mean of three wells ± s.e.m.

**Preparation and characterization of liposomal Wnt3a.** For preparation of liposomes, DMPC, DMPG (both from Lipoid AG) and cholesterol (Sigma) were mixed at molar ratios indicated in brackets, and dissolved in a 9/1 (v/v) mixture of chloroform/methanol. The solvent was then evaporated under vacuum on a rotavapor to generate a lipid film. The residual organic solvent was removed by a nitrogen flush. Following hydration of the lipid film in HEPES-buffered saline (HBS: 10 mM HEPES buffer (pH 7.2), 0.8% NaCl) or PBS, the lipid suspension was extruded 10 times each through 200 and 100 nm pore size polycarbonate filters (Northern Lipids) using nitrogen pressure and a Lipex high pressure extruder (Northern Lipids). The mean particle size and size distribution (polydispersity index) of the liposomes were determined with dynamic light scattering using a Malvern Zetasizer. The liposomes were stored under argon at 4 °C until use.

Purified Wnt3a protein (see above) was mixed with liposomes at a 1:7.5 ratio for final concentrations of 7–15 µg ml$^{-1}$ Wnt3a and 15 mM phospholipid, unless indicated otherwise. The Wnt3a liposomes were incubated for 1 h at 4 °C, followed by dialysis in HBS/PBS at 4 °C using a 10 kDa molecular weight cutoff membrane. To obtain higher Wnt3a concentrations (15–25 µg ml$^{-1}$) in the liposomes, multiple rounds of incubation with purified Wnt3a and subsequent dialysis were performed. Phospholipid concentrations were determined by a colorimetric phosphate assay[30]. First, 2–6 µl of liposome dispersions were dried in glass test tubes for 30 min at 180 °C. Phospholipids were then degraded by the addition of 0.3 ml 70% perchloric acid, followed by a 45 min incubation at 180 °C until the samples became colourless. Evaporation of the perchloric acid was prevented by placing porcelain marbles on top of the tubes. Once the test tubes were cooled down to room temperature, 1 ml of water was added, followed by 0.5 ml of 1.2% hexa-ammoniummolybdate solution. Samples were mixed by vortexing, 0.5 ml of a freshly prepared solution of 5% (w/v) ascorbic acid was added and vortexed again. Samples were then placed in boiling water for 5 min and subsequently allowed to cool down to room temperature. Samples were transferred to disposable cuvettes and absorbance was measured at 797 nm against a calibration curve prepared with known amounts of sodium phosphate (NaH$_2$PO$_4$).

To separate liposome-associated from -unincorporated Wnt3a proteins, the Wnt3a liposome suspensions were centrifuged for 30 min at 100,000g at 4 °C. The pellet was resuspended in PBS and together with the supernatant analysed by western blotting. Wnt3a protein was detected by a rabbit monoclonal anti-Wnt3a antibody (Cell Signaling Technology; 2721) using a goat anti-rabbit IRDye 800CW secondary antibody (Li-Cor) and imaged on an Odyssey Infrared Imaging System (Li-Cor). For each band, background-subtracted quantification numbers, the so-called integrated intensities, were generated with the analysis software provided. Ratios of integrated intensities were calculated and results were plotted as the percentage of either free or liposome-associated Wnt3a relative to the total amount.

Residual CHAPS content of Wnt3a reagents was determined by means of HPLC (Alliance Waters 2695, Waters, USA), using reversed-phase chromatography and UV detection (Dual λ Absorbance detector, Waters, USA) at a wavelength of 210 nm. The column used was a LiChrospher 100, RP-18 (5 µM). As a mobile phase, 4% acetonitrile, 95.9% water and 0.1% perchloric acid was used at a flow rate of 1.0 ml min$^{-1}$.

**Lyophilization of liposomal Wnt3a.** Liposomes (DMPC/DMPG/cholesterol 10:1:10) were prepared as described above, except that 10% sucrose in 10 mM HEPES (pH 7.4) was used instead of HBS for hydration of the lipid film. Wnt3a and vehicle liposomes were prepared as described above and dialysed in 10% sucrose in 10 mM HEPES (pH 7.4) at 4 °C. The liposomes were then freeze-dried in 200 µl aliquots in 3.5 ml flat bottom vials in a Lyovac GT4 freeze-dryer (Leybold-Heraeus, Cologne, Germany). The vials were placed on the freeze-dryer plate at a temperature of −40 °C. First, the plate temperature was maintained at −40 °C at a chamber pressure of 10–13 Pa for 24 h. Then, the plate temperature was increased to 0 °C and the chamber pressure was adjusted to 0.9–1 Pa. Twenty-four hours later, the chamber was filled with nitrogen gas, the vials closed and sealed with rubber and aluminium caps and stored at −20 °C until use. One hour before use, freeze-dried liposomes were allowed to come to room temperature and were reconstituted by adding 200 µl distilled water, followed by 1 min of rigorous vortexing. Particle size measurements were carried out using a Zetasizer (Malvern).

**Preparation of Wnt3a-coated PLGA NPs.** For preparation of lipid-coated PLGA NPs, we first prepared PLGA NPs using an oil/water emulsion and solvent evaporation–extraction method. In brief, for each preparation 100 mg of PLGA (Resomer RG 502 H, lactide:glycolide molar ratio 48:52 to 52:48; Sigma-Aldrich) in 3 ml of dichloromethane (DCM; Sigma-Aldrich) was added drop wise to 25 ml of aqueous 2% (w/v) polyvinyl alcohol (Sigma-Aldrich) in distilled water, and emulsified for 90 s using a Branson sonifier 250 sonicator. Next, a film of DMPC was prepared by dissolving 10 mg of DMPC in DCM, followed by evaporation of the DCM by a stream of nitrogen gas. Subsequently, the PLGA emulsion was rapidly added to the vial containing the lipids and the solution was homogenized for 30 s using a sonicator. Following overnight evaporation of the solvent at 4 °C, the lipid-coated PLGA NPs were collected by centrifugation at 10,000g for 10 min, washed three times with distilled water and lyophilized. The Z-average size and polydispersity index of the lipid-coated PLGA NPs were measured by dynamic light scattering using a Nano ZS Zetasizer (Malvern). The corresponding particle diameter was calculated assuming that the particles were spherical with a value of 178 ± 4 nm, while the polydispersity index with a value of 0.09 ± 0.01. To associate PLGA NPs with Wnt3a, the NPs were resuspended in PBS at a concentration of 10 mg ml$^{-1}$, incubated for 1 h with Wnt3a proteins at a 1:7.5 ratio for a final Wnt3a concentration of 7–10 µg ml$^{-1}$, followed by dialysis against PBS to remove CHAPS.

**Wnt activity assays.** Mouse LSL cells[31] (gift from Roel Nusse, Stanford University), expressing luciferase in response to TCF promoter binding, were cultured at 37 °C and 5% CO$_2$ in DMEM containing 10% FCS, 100 U ml$^{-1}$ penicillin and 100 µg ml$^{-1}$ streptomycin. The cells were tested for mycoplasma contamination every 2–3 months. For the activity assays, 25,000 LSL cells were plated in each well of a 96-well plate 24 h in advance. Wnt3a reagents and human recombinant afamin (Sino Biological; 13231-H08H-50) as indicated were separately incubated in serum-free culture medium for various periods of time at 37 °C in 96-well plates. On completion of incubation intervals, media containing Wnt3a reagents were added to LSL cells and incubated overnight, followed by cell lysis and luciferase activity assay using Promega luciferase assay reagent and a Glomax multiplate reader.

**Embryonic stem cell culture and clonal self-renewal assays.** R1 embryonic stem cells (obtained from Stanford Transgenic Facility) and R1-7xTcf-eGFP ESCs[16] were cultured on gelatine and FCS-coated plates in N2B27 medium, composed of one volume of DMEM/F12 and one volume neurobasal medium supplemented with 0.5% N2 supplement, 1% B27 supplement, 0.033% BSA 7.5% solution, 50 β-mercaptoethanol, 2 mM glutamax, 100 U ml$^{-1}$ penicillin and 100 µg ml$^{-1}$ streptomycin (all from Invitrogen). Purified Wnt3a of 250 ng ml$^{-1}$ was added to the ESC medium, and the medium was refreshed every day. ESCs were passaged at a ratio of 1:10 every 3 days as a single-cell suspension using 0.25% trypsin-EDTA and trypsin was quenched using soybean trypsin inhibitor (Sigma). The cells were tested for mycoplasma contamination every 2–3 months.

For assessment of Wnt-responsive reporter activation over several days, 50,000 R1-7xTcf-eGFP ESCs were plated on gelatine and FCS-coated 6-well plates in quadruplicates in the presence of 2 mM IWP2 (Merck; 681671). The same concentration of Wnt3a (250 ng ml$^{-1}$) either in the form of purified or liposomal Wnt3a as well as control lipid-vesicles containing the same amount of liposomes were added to the wells at day 0 only and the cells were no longer refreshed. The ESCs were imaged every day with an Olympus IX-70 inverted fluorescent microscope. Following imaging, one well of ESCs from each condition was analysed for GFP expression by flow cytometry (Fortessa, BD biosciences).

To quantify self-renewal of ESCs, single cells were plated at a clonal density of 200 cells per cm$^2$ in gelatine- and FCS-coated 6-well plates and in 24-well plates in triplicates. Media containing the indicated supplements were refreshed daily or each passage, as indicated. Every 3 days, 6-well plates were trypsinized to single cells, and passaged at a dilution that would result in clonal density again. Concurrently, the 24-well plates were stained for alkaline phosphatase by SCR004 Kit (Millipore) according to the manufacturer's instructions. Stained plates were rinsed with water, dried and counted manually under microscope. The cumulative number of colonies was determined by multiplying the colony counts by the dilution factor used for passaging. Results were plotted as the mean of three wells ± s.e.m.

**Quantitative RT–PCR analysis.** Organoid cultures were collected in RLT buffer from RNeasy Mini Kit (Qiagen) and RNA isolated according to the manufacturer's instructions. Reverse transcription was performed using Superscript II (Invitrogen). cDNA was amplified in triplicate with LightCycler 480 SYBR Green Master mix (Roche) on a Roche Lightcycler 480. Relative quantification was achieved by normalizing mean values to the GAPDH gene and reported ± s.e.m. The following primers were used: LGR5 forward, 5′-AGGTCTGGTGTGTTGCTGAG-3′ and LGR5 reverse, 5′-GTGAAGACGCTGAGGTTGGA-3′; TROY forward, 5′-AACT GTGTTCCCTGCAACCA-3′ and TROY reverse, 5′-GTCCTCCTTGAACCTGTGCA-3′.

**Histology and imaging.** For immunochemistry staining, organoids were fixed with 4% paraformaldehyde for 1 h at room temperature, embedded in paraffin, sectioned at 5 μm and sections processed for periodic acid-Schiff and alkaline phosphatase staining or immunohistochemical staining. Antibodies used were as follows: mouse anti-Ki67 (1:250, Monosan MONX10283), anti-chromogranin A (1:100; Santa Cruz; sc-1488) and anti-EphB2 (R&D systems; AF647). For whole-mount immunostaining, samples were fixed by the addition of equal amounts of 4% prewarmed paraformaldehyde to the wells and incubated for 2 h at 37 °C. The wells were washed three times with prewarmed PBS. Next, Matrigel was dissolved by the addition of cold Cell Recovery Solution (BD Biosciences), followed by a 30 min incubation at 4 °C. The recovered organoids were cytospun for 2 min at 500 r.p.m. on glass slides and antigens retrieved by boiling the samples in sodium citrate buffer (10 mM, pH 6.0) for 10 min. The samples were then washed three times for 10 min with PBS containing 0.2% Tween-20 (PBST), permeabilized with 0.1% Triton in PBS for 10 min at room temperature and blocked with 5% non-fat dry milk (Sigma) and 0.1% Triton in PBS for 30 min. Incubation with primary or secondary antibodies diluted in blocking solution was done overnight at 4 °C or 1–2 h at room temperature, respectively, followed by three washes in PBST for 10 min. The primary antibodies used were as follows: anti-Mucin2 (1:500, cat. no.: SC-15334; Santa Cruz), anti-lysozyme (1:1,000; cat. no.: A0099; Dako), anti-Villin (1:100; cat. no.: 610359; BD Biosciences) and anti-chromogranin A (1:250; cat. no.: NB120-15160; Novus Biological). The secondary antibodies were Alexa-Fluor-488 and Alexa-Fluor-594. Nuclei were stained with DAPI (Molecular Probes) for 5 min at room temperature. Immunofluorescence confocal images were acquired using a Zeiss LSM 700 inverted laser-scanning confocal microscope equipped with an external argon laser.

**Statistical analysis.** All luciferase reporter, colony counting and RT–PCR assays were performed as three technical replicates. Standard errors of the means were calculated assuming normal distribution of the data.

**Data availability.** All relevant data generated or analysed during this study are included in this published article and its Supplementary Information Files or from the corresponding author on reasonable request.

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

## Acknowledgements

We thank Patrick Franken and Ricardo Fodde for their help with immunostainings, and Gerben Koning (deceased December 2015) and Wouter Lokerse for assistance with liposome production. This study was supported by grants from TI Pharma (D5-402), the Netherlands Institute for Regenerative Medicine (FES0908) and ZonMw (116006104).

## Author contributions

Conceived and designed experiments: N.T., M.M.A.V., L.J.W.v.d.L., R.V., E.B., E.M., J.J.C., H.C. and d.t.B. Performed the experiments: N.T., L.v.B., S.v.d.B., H.B. and M.M.A.V. Analysed the data: N.T., E.B., E.M., J.J.C. and d.t.B. Contributed reagents or materials: L.J.C. and J.d.J. Wrote the paper: N.T., L.H., E.B., E.M., J.J.C. and d.t.B.
