## [Peer Review File · Nature Communications]

Reviewer #2 (Remarks to the Author)

This manuscript describes the use of a liposomal Wnt3a formulation in the derivation and growth of primary intestinal organoid cultures. The data, which are of high quality and convincing, show that this liposomal Wnt formulation is more stable over time relative to purified Wnt protein. Furthermore, liposomal Wnt3a overcomes the cytotoxic effects associated with the detergent CHAPS, which is present in purified Wnt3a. The increased stability and shelf life of these Wnt liposomes represent an important technical advance in using Wnt proteins in stem cell cultures.

Major comments:

This work represents primarily a technical advancement, which is certainly important and valuable. However, the impact of this work would be significantly enhanced by expanding on these current findings. For example, the authors provide data in Figure 6 that shows that this Wnt formulation can be used to establish and expand organoids from human jejunum and liver. Does purified Wnt3a fail to establish such cultures? It would also be interesting to extend these findings to other Wnts, for example Wnt5a? And, can the use of the method shown in Figure 5d of extracting Wnt3a from crude a Wnt preparation be extended to other Wnts that have eluded purification attempts?

As presented, it is unclear to what extent this liposomal Wnt3a formulation is an improvement over other formulations. Many factors have been shown to potentiate and/or stabilize Wnt3a activities, including Sfrp1 (Xavier et al. Cell Signal 26:94-101, 2014), HSPGs (Fuerer et al. Developmental Dynamics 239:184-190, 2010), Afamin (Mihara et al., Elife 2016), BSA (present in some commercially available preparations because it "enhances protein stability, increases shelf-life, and allows the recombinant protein to be stored at a more dilute concentration" [source R&D website]), to name a few. The authors should provide some experiments that address whether their Wnt formulation is an improvement over these other methods.

The experiments shown in Figure 4 demonstrate that addition of liposomes stabilizes purified Wnt3a protein to a similar extent as liposomal Wnt3a formulations do. This raises the distinct possibility that one mode of action of these liposomes is to detoxify the purified Wnt3a preparation by soaking up the cytotoxic CHAPS. Therefore, is it possible that in this setting liposomes act as "detergent buffers"? To address this point, the authors could compare their liposome formulation to some of the commercially available lipid-rich formulations commonly used in certain serum free culture media.

Minor comments

Figure 1a: have the authors tested Wnt3a concentrations lower than 400 ng/ml? Perhaps lower concentrations are more effective than higher concentrations.

Figure 1a: the effect of high cholesterol in stabilizing Wnt3a activity is impressive. Could the authors show a more extensive dose response; as is, there is very little difference of cholesterol from 0 to 4. Only at 10 does cholesterol exert a dramatically positive effect on Wnt3a activity. What about at molar ratios between 4 and 10? and above 10?

Experiments clearly show that liposomal Wnt3a retains activity over time relative to purified Wnt3a. It would also be interesting to know whether liposomal Wnt3a is more active than purified Wnt3a. A dose response experiment could potentially make the point that liposomal Wnt3a has higher specific activity than purified protein.

The dose-dependency of liposomal Wnt3a supporting organoid formation (line 93-94; Figure 3a) is weak: the authors show three concentrations: 400 and 800 ng/ml are essentially the same, while 1.5 ug/ml shows a clear benefit. To make a point of a dose dependent effect the authors should show the effect of additional concentrations.

Can GSK3 inhibitors be used in place of Wnt3a formulations to establish, maintain and passage intestinal organoids? While there are clear advantages of using Wnt formulations, such as these liposomes, the authors should discuss briefly the potential use of GSK3 inhibitors and its pitfalls/shortcomings, or better, perform an experiment that compares a GSK3 inhibitor to Wnt liposomes.

Figure 5d: the ability of liposomes to extract Wnt3a from crude Wnt3a preparations is quite impressive. Additional characterization of this method would be desirable: for example, do liposomes deplete the Wnt3a protein from the crude protein sample? What is the approximate binding capacity of these liposomes for Wnt3a?

Abstract: rephrase first sentence "..., but are only effective as a serum-containing, conditioned medium." Many papers show experiments that used commercially available Wnt proteins, presumably with some success. The authors are likely correct in their argument that these Wnt protein preparations are sub-optimal, however, their point in the first sentence is over-stated and should be softened (e.g. "..., however, most Wnt protein formulations are poorly active and highly labile.")

Regarding this statement: "We previously observed that Wnt proteins associate with lipid vesicles, which prolongs their activity^{14, 15}." Reference 14 is from another research group, therefore, this sentence should start with "We and others..."

Reviewer #4 (Remarks to the Author)

Wnt signalling proteins are an essential component for culturing human-derived organoids, but the investigators show that these are only effective as a serum-containing, conditioned medium. This is a limitation for the culture of organoids. The study demonstrates that Wnt3a activity is stabilized in culture by using lipid carriers. Stabilized Wnt3a supports increased self-renewal of organ and embryonic stem cells in serum-free conditions for establishment of healthy and diseased human intestinal and liver organoids. The development of these conditions is important because conditioned medium (that is currently used) contains undefined, differentiation-inducing components that may be undesirable for diagnostic assays. The findings are novel. The data presented is convincing and supports the conclusions of the research. The major finding is that liposomes facilitate the production and stabilization of high purity Wnt3a protein. I have the following concern:

Regarding Figures 3 and 6: There is no quantification for the efficiency of organoid development with each passage in the human duodenum, jejunum and liver organoids.

Reviewer #5 (Remarks to the Author)

I have looked at the rebuttal and the manuscript and believe that the authors satisfactorily addressed the concerns. The manuscript describes a method that will be useful for the growing community of labs that use organoid culture as a proxy for patient tissues and also adds the knowledge how Wnt proteins can be kept in an active form. This question is not new but still largely unresolved.

The concern about contamination by serum proteins is valid but I feel can be addressed by a more critical discussion in the manuscript about the limitation of the current method and potential future directions. For ex vivo experiments – as described here - this is not a major concern

Response to referees comments

We wish to thank the reviewers for their time and dedication to reviewing our work. Thanks to their insightful comments we have been able to greatly improve our manuscript, strengthen it in multiple areas, and place it in the context of existing work on stabilizing Wnt and growing human organoids. We believe that the revised manuscript will be a worthwhile addition to the rapidly growing field of human organoid research.

Reviewer #2:

This manuscript describes the use of a liposomal Wnt3a formulation in the derivation and growth of primary intestinal organoid cultures. The data, which are of high quality and convincing, show that this liposomal Wnt formulation is more stable over time relative to purified Wnt protein. Furthermore, liposomal Wnt3a overcomes the cytotoxic effects associated with the detergent CHAPS, which is present in purified Wnt3a. The increased stability and shelf life of these Wnt liposomes represent an important technical advance in using Wnt proteins in stem cell cultures.

We thank the reviewer for the careful assessment of our manuscript. We are also very grateful to the reviewer for highlighting new directions for further investigation and pointing out additional applications for our findings. We hope we will be able to realize some of these suggestions in the near future.

Major comments:

This work represents primarily a technical advancement, which is certainly important and valuable. However, the impact of this work would be significantly enhanced by expanding on these current findings.

1) For example, the authors provide data in Figure 6 that shows that this Wnt formulation can be used to establish and expand organoids from human jejunum and liver. Does purified Wnt3a fail to establish such cultures?

Purified Wnt3a was indeed unable to support establishment of jejunum cultures. This data was provided in **Table 1**, and we now show additional data in **Supplementary Figure 2a**.

For liver this was not tested. The question we sought to address at this stage was whether Wnt3a liposomes would support the derivation of stem cell cultures from other human organs. We were fortunate that our collaborators at the department of surgery were willing to test Wnt3a liposomes alongside the established condition, Wnt3a conditioned medium, for the derivation of human liver organoids. Their primary aim was to establish organoids from donor

material to study liver disease and they were willing to test Wnt3a liposomes because of our good results with jejunum. The results demonstrate that Wnt3a liposomes can substitute for Wnt3a conditioned medium in the derivation of human liver organoids, suggesting that Wnt3a liposomes will be applicable in a wide variety of applications now relying on Wnt3a conditioned media.

2) It would also be interesting to extend these findings to other Wnts, for example Wnt5a?

The growth factors required to support human organoids had been identified earlier (see e.g. Sato et al 2011 Gastroenterology 141, 1762; Huch et al 2015 Cell 160, 299) and specifically include Wnt3a, a Wnt ligand that activates the canonical beta-catenin pathway. Wnt5a is a different category of Wnt ligands that generally does not activate the canonical/beta-catenin pathway (see e.g. Oishi et al 2003 Genes Cells 8, 645), and we are not aware of clinically relevant applications for purified Wnt5a that would require serum-free media. In fact, Wnt5a is sometimes even used as a negative control for Wnt3a (see e.g. Xu et al 2016 PNAS 113, E6382). Moreover, since Wnt5a does not activate the beta-catenin pathway we do not have an assay for its activity. It is therefore difficult to establish whether its stability is increased by association with liposomes. For these reasons we have focused on stabilizing Wnt3a.

3) And, can the use of the method shown in Figure 5d of extracting Wnt3a from crude a Wnt preparation be extended to other Wnts that have eluded purification attempts?

We thank the reviewer for the interesting suggestion. The problem with other Wnts in our experience is primarily that it is almost impossible to obtain any significant amount of them, probably because even overexpressing cells simply secrete them in extremely tiny amounts (Brown et al 1987 Mol Cell Biol 7, 3971; Papkoff et al 1987 Mol Cell Biol 7, 3978). We have in the past attempted purification of other Wnts, e.g. Wnt1, but found that, even though some cell lines would initially produce reasonable amounts of protein, they would do so only briefly. Perhaps the Wnt protein has an impact on the producer cells themselves and is the problem more likely addressed by manipulating the producer cells rather than using liposomes, although liposomes may be useful once the problems with secretion have been solved. Perhaps addition of liposomes to the producer cell culture may promote release of the Wnt protein in the medium. We have not yet explored this interesting possibility because it introduces additional complications, such as adapting the producer cells to serum-free media.

4) As presented, it is unclear to what extent this liposomal Wnt3a formulation is an improvement over other formulations. Many factors have been shown to potentiate

and/or stabilize Wnt3a activities, including Sfrp1 (Xavier et al. Cell Signal 26:94-101, 2014), HSPGs (Fuerer et al. Developmental Dynamics 239:184-190, 2010), Afamin (Mihara et al., Elife 2016), BSA (present in some commercially available preparations because it “enhances protein stability, increases shelf-life, and allows the recombinant protein to be stored at a more dilute concentration” [source R&D website]), to name a few. The authors should provide some experiments that address whether their Wnt formulation is an improvement over these other methods.

The reviewer raises a very good point. We have tested several of these other methods but found that they ultimately provided little benefit to Wnt stability:

- Sfrp1**: We have tested the related molecule Fz8CRD (Hsieh et al 1999 PNAS 96, 3546). We found however that this acted as a competitive inhibitor of Wnt signaling (see e.g. ten Berge et al 2008 Development 135, 3247). Moreover, Sfrp1 was shown to potentiate the Wnt pathway via interactions with the Frizzled receptor, not by stabilizing the Wnt ligand (Xavier et al 2014 Cell Signal 26, 94).
- HSPGs**: These cost €700 per 100 µg (Sigma H4777) and were reportedly used at a concentration of 10 µg/ml (Fuerer et al 2010 Dev Dyn 239, 184), adding €70,000 to the cost of a liter of medium. This is clearly not economical. Moreover, HSPGs are already present at high concentrations in the Matrigel present in the organoid cultures (Kibbey 1994 J Tissue Culture Methods 16, 227).
- Afamin**: We have now tested recombinant afamin but we did not observe an effect on the stability of Wnt3a protein, even when present at concentrations found in serum. Perhaps the stabilization of Wnt by afamin occurs only when the molecules are complexed during their biosynthesis. This data has now been added to the manuscript (**Supplementary Figure 3; page 8 first paragraph**).
- BSA**: The effect of BSA on Wnt3a stability was already tested by Fuerer et al, who found that it did not increase Wnt3a stability in serum-free medium (Fuerer et al 2010 Dev Dyn 239, 184). Moreover, BSA was already present in our Wnt3a preparations (see **Figure 5a,d**). The statement by R&D is a standard statement added to most recombinant proteins in their inventory (except carrier-free versions) and does not indicate that they actually tested the effect of BSA on Wnt3a stability in serum-free cell culture conditions.

5) The experiments shown in Figure 4 demonstrate that addition of liposomes stabilizes purified Wnt3a protein to a similar extent as liposomal Wnt3a formulations do. This raises the distinct possibility that one mode of action of these liposomes is to detoxify the purified Wnt3a preparation by soaking up the cytotoxic CHAPS. Therefore, is it possible that in this setting liposomes act as “detergent buffers”? To address this point, the authors could compare their liposome formulation to some of the commercially available lipid-rich formulations commonly used in certain serum free culture media.

This is a very intriguing suggestion; if liposomes would absorb CHAPS this may indeed contribute to increased stem cell self-renewal. Instead of using commercially available lipid-rich formulations, for which we do not know how they affect Wnt stability, we have investigated this using liposomes and our embryonic stem cell (ESC) model. Regular Wnt3a at 250 ng/ml (the same concentration as used in **Figure 4a**) is sufficient to maintain ESCs as long as the Wnt3a is replaced daily and endogenous Wnt production is not inhibited (see ten Berge et al 2011 Nat Cell Biol 13, 1070, figure 4). We observed that in these conditions the regular Wnt3a performed similar to Wnt liposomes, see **chart below**. This indicates that the level of CHAPS present at this concentration of Wnt3a is not impacting ESC self-renewal. When Wnt3a or Wnt3a liposomes are added once per passage, ESC self-renewal is strongly improved by the liposomes (**Figure 2d**), indicating that it is the extended stability of the Wnt liposomes that is responsible for their enhanced efficacy.

Minor comments

6) *Figure 1a: have the authors tested Wnt3a concentrations lower than 400 ng/ml? Perhaps lower concentrations are more effective than higher concentrations.*

Yes, we tested jejunum samples with lower, and higher, concentrations of Wnt3a. We obtained organoids from 1 out of 1 sample at 1,000 ng/ml Wnt3a liposomes, 3 out of 3 samples using 800 ng/ml, 2 out of 3 with 400 ng/ml, and 0 out of 1 using 200 ng/ml. This data is now added to **Table 1**, and additional quantification is shown in **Supplementary Figure 2a (page 8 last paragraph)**. Collectively, the data does not suggest that lower Wnt3a concentrations would be more effective.

7) *Figure 1a: the effect of high cholesterol in stabilizing Wnt3a activity is impressive. Could the authors show a more extensive dose response; as is, there is*

very little difference of cholesterol from 0 to 4. Only at 10 does cholesterol exert a dramatically positive effect on Wnt3a activity. What about at molar ratios between 4 and 10? and above 10?

We assume the reviewer is referring to figure 2a. The difference between 1:4 and 1:10 ratios of cholesterol is indeed large and repeatedly observed. While it is certainly possible that the composition of the liposomes could be further fine-tuned by making smaller changes in the lipid ratios, or by adding other lipids, we show that the current composition works very effectively. We feel that further fine-tuning would not substantially advance our insight and is more appropriately done in the context of a company setting with a view of improving physical stability, shelf life, scalability and cost, leading to a commercially available product. Furthermore, the cholesterol concentration cannot be raised further because it is close to its solubility limit in DMPC liposomes (Huang et al 1999 BBA 1417, 89).

8) Experiments clearly show that liposomal Wnt3a retains activity over time relative to purified Wnt3a. It would also be interesting to know whether liposomal Wnt3a is more active than purified Wnt3a. A dose response experiment could potentially make the point that liposomal Wnt3a has higher specific activity than purified protein.

We have now added a dose-response experiment to the manuscript. The results indicate that there is no difference in specific activity between regular purified Wnt3a and Wnt3a liposomes (**Supplementary Figure 1c; page 5 first paragraph**).

9) The dose-dependency of liposomal Wnt3a supporting organoid formation (line 93-94; Figure 3a) is weak: the authors show three concentrations: 400 and 800 ng/ml are essentially the same, while 1.5 ug/ml shows a clear benefit. To make a point of a dose dependent effect the authors should show the effect of additional concentrations.

The data, which was already presented in the same form in the original manuscript, shows that expansion in 800 ng/ml is improved 1.42-fold compared to 400 ng/ml (**Figure 3a**). Perhaps the magnitude of the improvement is not immediately obvious because of the logarithmic scale in which it is plotted. In the revised manuscript we now show the expansion of jejunum organoids in response to different Wnt liposome concentrations, where we observe perhaps a clearer dose-dependent response (**Supplementary Figure 2a**). We feel that the data quite strongly demonstrates the benefit of higher Wnt3a concentrations.

10) *Can GSK3 inhibitors be used in place of Wnt3a formulations to establish, maintain and passage intestinal organoids? While there are clear advantages of using Wnt formulations, such as these liposomes, the authors should discuss briefly the potential use of GSK3 inhibitors and its pitfalls/shortcomings, or better, perform an experiment that compares a GSK3 inhibitor to Wnt liposomes.*

This is a very good question: GSK3 inhibitors such as CHIR99021 are far cheaper, more stable and easier to use than Wnt ligands, and have been used successfully with mouse intestinal stem cells (Yin et al 2014 Nature Methods 11, 106). We were however unable to propagate human organoids using CHIR99021. This is now shown in **Supplementary Figure 4 (page 8 first paragraph)**. This same CHIR99021 is used successfully in our lab to culture mouse embryonic stem cells, verifying its activity. It should be noted that also with other human adult stem cells Wnt signals can often not be replaced by GSK3 inhibitors (e.g. Narcisi et al 2016 Tissue Eng. Part A. doi:10.1089/ten.TEA.2016.0081).

11) *Figure 5d: the ability of liposomes to extract Wnt3a from crude Wnt3a preparations is quite impressive. Additional characterization of this method would be desirable: for example, do liposomes deplete the Wnt3a protein from the crude protein sample?*

We show in **Supplemental Figure 1b (page 8 first paragraph)** that the liposomes extract approximately 85% of the total amount of Wnt3a from the crude sample.

12) *What is the approximate binding capacity of these liposomes for Wnt3a?*

We calculated a theoretical upper level of the number of Wnt molecules that could bind to the liposomes using some known numbers and reasonable approximations:

The phospholipid concentration is 15 mM and the Wnt3a concentration is approximately 10 mg/l.

The mol weight of Wnt3a is 39 kDa

N_A is Avogadro's number

The total number of lipid molecules in a 100 nm DMPC liposome is approximately 80,000 (see e.g. <http://www.liposomes.org/2009/01/number-of-lipid-molecules-per-liposome.html>).

From this we calculate that we have approximately $15 \text{ mM} \times N_A / 80,000 = 10^{17}$ liposomes per liter and $(10 \text{ mg} \times N_A) / 39 \text{ kDa} = 1.5 \times 10^{17}$ molecules of Wnt3a/l.

Thus in our current experiment we had a ratio of approximately 1 Wnt3a molecule per liposome.

The surface area of the Wnt molecule when bound via its lipid is approximately 30 nm² (Janda et al 2012 Science 337, 59).

The surface area of a 100 nm liposome is $4\pi 2500\text{nm}^2=31,400\text{ nm}^2$

To cover an entire 100 nm liposome with Wnt molecules would therefore require approximately 1,000 Wnt molecules.

These calculations indicate that the liposomes are far from saturated and could extract perhaps a 100-fold or even larger amount of Wnt3a protein from the sample than present in our current experiment. We do not have the experimental means to achieve such a high concentration of Wnt3a protein. However, these calculations suggest that saturation of the liposomes with Wnt3a protein is unlikely to limit their performance.

13) Abstract: rephrase first sentence "..., but are only effective as a serum-containing, conditioned medium." Many papers show experiments that used commercially available Wnt proteins, presumably with some success. The authors are likely correct in their argument that these Wnt protein preparations are sub-optimal, however, their point in the first sentence is over-stated and should be softened (e.g. "..., however, most Wnt protein formulations are poorly active and highly labile.")

We adjusted the abstract according to the reviewer's suggestion.

14) Regarding this statement: "We previously observed that Wnt proteins associate with lipid vesicles, which prolongs their activity^{14, 15}." Reference 14 is from another research group, therefore, this sentence should start with "We and others..."

Both papers are from the same research group with which we collaborated closely and co-developed Wnt liposomes with in the past. However, we appreciate the reviewer's point and modified the manuscript according to the reviewer's suggestion.

Reviewer #4:

Wnt signalling proteins are an essential component for culturing human-derived organoids, but the investigators show that these are only effective as a serum-containing, conditioned medium. This is a limitation for the culture of organoids. The study demonstrates that Wnt3a activity is stabilized in culture by using lipid carriers. Stabilized Wnt3a supports increased self-renewal of organ and embryonic stem cells in serum-free conditions for establishment of healthy and diseased human intestinal and liver organoids. The development of these conditions is important because conditioned medium (that is currently used) contains

undefined, differentiation-inducing components that may be undesirable for diagnostic assays. The findings are novel. The data presented is convincing and supports the conclusions of the research. The major finding is that liposomes facilitate the production and stabilization of high purity Wnt3a protein. I have the following concern:

Regarding Figures 3 and 6: There is no quantification for the efficiency of organoid development with each passage in the human duodenum, jejunum and liver organoids.

We thank the reviewer for the succinct summary and positive evaluation of our work. With regards to the specific comment, we assume that the reviewer is concerned that the organoids expand poorly in Wnt3a liposomes, even though the liposomes support their initial establishment. Many labs indeed struggle with the culture of human organoids in Wnt3a conditioned medium. Quantifications of the expansion of duodenum organoids in response to different doses of Wnt3a liposomes were shown in **Figure 3a** (passage 7-8), **Figure 4b** (passage 10-11), and **Figure 5c** (passage 9-10), demonstrating that Wnt3a liposomes support the long-term expansion of the organoids. We have now specified the passage numbers in the figure legends to indicate this observation. In newly added data, where we tested the effect of GSK3 inhibition on duodenum organoid expansion in response to a request from reviewer 2, we show an additional quantification of duodenum organoid expansion at passage 12-13 (**Supplementary Figure 4**). Furthermore, the duodenum organoids were maintained for more than 6 months while passaged at a ratio of 1:6-1:8 every 7 to 10 days, in line with previously reported results (Sato et al 2011 Gastroenterology 141, 1762). We have added a statement to the manuscript detailing the robust long term expansion of the duodenum organoids (**page 5 last paragraph**).

With regards to jejunum organoids, we obtained an approximately 6-fold expansion per passage in the presence of Wnt3a liposomes, which is now shown for passages 6-8 in the new **Supplementary Figure 2b**. Only few organoids developed when using regular purified Wn3a or Wnt3a conditioned medium, and these did not expand upon passaging (**Table 1, Supplementary Figure 2a**). Like the duodenum organoids, the jejunum organoids were maintained for more than 3 months while passaged at a ratio of 1:6-1:8 every 7 to 10 days, in line with previously reported results (Sato et al 2011 Gastroenterology 141, 1762). Thus, also the jejunum organoids displayed robust expansion (**page 8 last paragraph**).

With regards to liver organoids, these require added Wnt signals only during the first 3 days of derivation (Huch et al 2015 Cell 160, 299); thus organoid development following passaging is independent from exogenous Wnt signals.

Their expansion varied with donor and is likely influenced by the disease state. These are properties currently under investigation and fall outside the scope of this manuscript.

Reviewer #5:

I have looked at the rebuttal and the manuscript and believe that the authors satisfactorily addressed the concerns. The manuscript describes a method that will be useful for the growing community of labs that use organoid culture as a proxy for patient tissues and also adds the knowledge how Wnt proteins can be kept in an active form. This question is not new but still largely unresolved.

The concern about contamination by serum proteins is valid but I feel can be addressed by a more critical discussion in the manuscript about the limitation of the current method and potential future directions. For ex vivo experiments – as described here - this is not a major concern.

We thank the reviewer for the positive and constructive comments. We have expanded the discussion with a section about the use of recombinant Wnt proteins for therapeutic purposes (**page 7 last paragraph**). In essence, while ideally the Wnt would be produced serumfree, recombinant proteins produced in the presence of sera that are verified free from known infectious disease markers (e.g. bovine sera should be sourced from certified TSE/BSE-free herds) are compatible with clinical applications (see *Guidance for FDA Reviewers and Sponsors: Content and Review of Chemistry, Manufacturing, and Control (CMC) Information for Human Gene Therapy Investigational New Drug Applications (INDs)*) as long as the final concentration of the serum in the administered product does not exceed 1:1,000,000 (FDA 21 CFR 610.15(b)). The development of serum-free stabilizers to produce Wnt proteins or of alternative molecules that activate the Wnt pathway are potential avenues towards completely eliminating sera.

Reviewer #4 (Remarks to the Author)

I have evaluated the response to the reviewer comments and the manuscript. The authors satisfactorily addressed the concerns. In particular, the investigators have clearly detailed passage numbers and organoid efficiency for each organ system studied. I have no remaining concerns.